# Antimicrobial Activity of the Green Tea Polyphenol (−)-Epigallocatechin-3-Gallate (EGCG) against Clinical Isolates of Multidrug-Resistant *Vibrio cholerae*

**DOI:** 10.3390/antibiotics11040518

**Published:** 2022-04-13

**Authors:** Achiraya Siriphap, Anong Kiddee, Acharaporn Duangjai, Atchariya Yosboonruang, Grissana Pook-In, Surasak Saokaew, Orasa Sutheinkul, Anchalee Rawangkan

**Affiliations:** 1School of Medical Sciences, University of Phayao, Phayao 56000, Thailand; achiraya.si@up.ac.th (A.S.); anong.ki@up.ac.th (A.K.); achara.phso@gmail.com (A.D.); f.atchariya@hotmail.com (A.Y.); krissy_seven@hotmail.com (G.P.-I.); 2Unit of Excellence in Research and Product Development of Coffee, Division of Physiology, School of Medical Sciences, University of Phayao, Phayao 56000, Thailand; 3Department of Pharmaceutical Care, Division of Social and Administrative Pharmacy, School of Pharmaceutical Sciences, University of Phayao, Phayao 56000, Thailand; saokaew@gmail.com; 4Center of Health Outcomes Research and Therapeutic Safety (Cohorts), School of Pharmaceutical Sciences, University of Phayao, Phayao 56000, Thailand; 5Unit of Excellence on Clinical Outcomes Research and Integration (UNICORN), School of Pharmaceutical Sciences, University of Phayao, Phayao 56000, Thailand; 6Faculty of Public Health, Mahidol University, Bangkok 10400, Thailand; orasa.sut@mahidol.ac.th

**Keywords:** antimicrobial activity, EGCG, green tea, MDR, *Vibrio cholerae*

## Abstract

The spread of multidrug-resistant (MDR) *Vibrio cholerae* necessitates the development of novel prevention and treatment strategies. This study aims to evaluate the in vitro antibacterial activity of green tea polyphenol (−)-epigallocatechin-3-gallate (EGCG) against MDR *V. cholerae*. First, MIC and MBC values were evaluated by broth microdilution techniques against 45 *V. cholerae* strains. The checkerboard assay was then used to determine the synergistic effect of EGCG and tetracycline. The pharmaceutical mode of action of EGCG was clarified by time-killing kinetics and membrane disruption assay. Our results revealed that all of the 45 clinical isolates were susceptible to EGCG, with MIC and MBC values in the range of 62.5–250 µg/mL and 125–500 µg/mL, respectively. Furthermore, the combination of EGCG and tetracycline was greater than either treatment alone, with a fractional inhibitory concentration index (FICI) of 0.009 and 0.018 in the O1 and O139 representative serotypes, respectively. Time-killing kinetics analysis suggested that EGCG had bactericidal activity for MDR *V. cholerae* after exposure to at least 62.5 µg/mL EGCG within 1 h. The mode of action of EGCG might be associated with membrane disrupting permeability, as confirmed by scanning electron microscopy. This is the first indication that EGCG is a viable anti-MDR *V. cholerae* treatment.

## 1. Introduction

Cholera is an acute diarrhoeal infection caused by consuming contaminated food or water containing *Vibrio cholerae*, a gram-negative bacteria. *V. cholerae* infection can kill within hours if left untreated, especially with serogroups O1 and O139, which have the potential to cause cholera outbreaks throughout the world [1]. According to the most recent global burden estimate, there are approximately 1.3 to 4.0 million cholera cases per year, with 21,000 to 143,000 deaths worldwide [2]. Despite the availability of a vaccine, 923,037 cases were reported from 31 countries in 2019, with 1911 deaths (a mortality rate of 0.2%) [3]. As a consequence, the World Health Organization (WHO) aims to decrease cholera deaths by 90% by 2030 [4].

The primary treatment for *V. cholerae* was oral rehydration therapy in combination with antimicrobial agents, such as tetracycline, fluoroquinolones, and azithromycin [5]. Treatment failures have become more common in recent years as a result of the recurrence of antimicrobial-resistant *V. cholerae* [6,7,8,9,10]. The emergence of drug-resistant *V. cholerae* is a global health concern because the infections seem to be severe and difficult to treat. Infections with drug-resistant *V. cholerae* could lead to higher case fatality rates, longer hospitalizations, more secondary infections, and higher health-care costs in various countries, including Thailand [11]. Previously, we reported that 61.5% (48 of 78 isolates) of *V. cholerae* isolates between 1991 and 2013 were antimicrobial-resistant strains, with 56.3% of them being multidrug resistant (MDR), and conferring resistance to three or more antimicrobial classes [12]. It is important to note that the development of antibiotic resistance outpaces the development of new drugs, resulting in a global problem with long-term negative consequences. Therefore, the finding of alternative anti-Vibrio compounds, particularly those derived from plants, has become critical.

Green tea (*Camellia sinensis*) is a well-known natural source of polyphenols, including phenolic acids (caffeic acid and gallic acid) and flavonoids. Catechins from green tea belong to the family of flavonoids containing flavan-3-ol units and galloylated catechins. Green tea catechins are characterized by the presence of a benzopyran structure with at least one aromatic ring (Appendix A) [13,14]. Numerous studies have found that drinking green tea provides a wide range of health benefits, including antimicrobial activity against a variety of organisms [15,16,17]. 120 mL of green tea infusion contains approximately 150 mg of catechins, which includes 10–15% (−)-epigallocatechin gallate (EGCG), 6–10% (−)-epigallocatechin (EGC), 2–3% (−)-epicatechin gallate (ECG), and 2% (−)-epicatechin gallate (EC) [18,19]. Green tea consumption has been shown to distribute these compounds and/or their metabolites throughout the body, allowing for not only the treatment of infection, but also its prevention [20]. EGCG is the most effective biological compound for anti-infective properties, i.e., against viruses, bacteria, and fungi [21,22,23]. Green tea catechins exhibit broad antibacterial activity against both gram-positive and gram-negative bacteria through a variety of mechanisms, including the inhibition of cell wall and cell membrane synthesis, protein and nucleic acid synthesis, or the inhibition of metabolic pathways, such as toxins and extracellular matrix virulence factors, oxidative stress, iron chelation, and so on [16,17,24]. For *V. cholerae*, green tea catechins reveal the inhibition of bacterial growth and/or cholera toxin secretion [25,26]. Moreover, an in vivo study indicated that catechins, especially EGCG at a concentration of 0.5 to 1.0 mg/mL, had anti-cholera activity by inhibiting cholera toxin-induced fluid accumulation in mice and preventing *V. cholerae* infection in rabbits, using the rabbit ileal loop model [27,28]. However, there have been few reports of data, and the mechanism remains unknown.

As a consequence, we shed light on the potential antimicrobial properties of EGCG against MDR *V. cholerae* by examining its antimicrobial activity and investigating its synergistic effects with the antibiotic tetracycline. Furthermore, we also assess the pharmacological mode of action of EGCG with respect to the potential disruption of the membrane of the microorganisms and its effect on bacterial morphology, which may be useful in bringing about a new opportunity in complementary and alternative medicine.

## 2. Results

### 2.1. EGCG Inhibits Drug Resistant V. cholerae Strains

We investigated the minimum inhibitory concentration (MIC) and minimum bactericidal concentration (MBC) of EGCG on the 45 *V. cholerae* clinical strains that maintained an antibiotic resistance pattern, such as streptomycin (STM), colistin (COL), nalidixic acid (NAL), sulfamethoxazole (SMX), tetracycline (TET), trimethoprim (TMP), ciprofloxacin (CIP), and azithromycin (AZI). The results, which are expressed as MIC and MBC values in Table 1, showed that all 45 isolates were sensitized to EGCG by expressed MIC values of 62.5 to 250 µg/mL, while MBC values were 125 to 500 µg/mL. The most effective against clinical strains had a MIC of 125 µg/mL (29/45), followed by 62.5 µg/mL (15/45), and 250 µg/mL (1/45), which corresponds to 64.44%, 33.33%, and 2.22%, respectively. Meanwhile, the bactericidal activity of EGCG showed MBC values of 250 µg/mL (29/45), followed by 125 µg/mL (15/45) and 500 µg/mL (1/45), as shown in Appendix A. On the other hand, the MIC and MBC values of tetracycline–a positive control antibiotic–for drug-resistant *V. cholerae* strains were shown to range from 0.48 to 62.5 µg/mL and 0.97 to 125 µg/mL, respectively (Appendix A). This suggests that the effects of EGCG on antimicrobial activities vary depending on the *V. cholerae* clinical strains.

Considering MDR *V. cholerae* as a potential threat to public health, we investigated the synergistic effect of EGCG and tetracycline, the WHO-recommended first-line drug choice for cholera treatment, in the representative MDR strains, P48 *V. cholerae* O1 El Tor Ogawa, as well as 22,136 *V. cholerae* O139 strains. It is important to note that the P48 O1 strain is a six-drug resistance strain (AZI, COL, NAL, SMX, TET, and TMP), whereas 22,136 O139 is a three-drug resistance strain (COL, SMX, TMP). Both serogroups were chosen as representative strains for future study because they carried the virulence genes up to 10 genes, i.e., *ctxA*, *ctxB*, *zot*, *ace*, *tcpA*, *hlyA*, *rtxA*, *ompU*, *toxR*, and *mshA*, which are highest among O1 and O139 strains [29]. Table 2 shows the evaluation of the synergistic effect of EGCG and tetracycline. In both standard and MDR strains, the combination effect on bacterial growth appeared to be greater than treatment alone (MIC 125 µg/mL), where the combination treatment reduced the MIC value to 0.97 µg/mL. It is important to note that the fractional inhibitory concentration index (FICI) is 0.009 for the O1 (P48) and reference strain, and 0.018 for the O139 (22136) strains. These results indicate that combining EGCG with tetracycline might be a more effective treatment for MDR *V. cholerae* than either treatment alone.

### 2.2. Analysis of Bacterial Killing Kinetics

We then investigated the time-kill kinetics of EGCG on the viability of *V. cholerae* O1 and O139 in order to define the bactericidal level, using a 0.25 to 4-fold MIC treatment. Figure 1 demonstrates the time-killing curve analysis. The kill kinetic profiles of 0.5× MIC (62.5 µg/mL), 1× MIC (125 µg/mL), 2× MIC (250 µg/mL), and 4× MIC (500 µg/mL) of EGCG displayed rapid bactericidal activity in all tested strains, with an approximate colony-forming units (CFU) reduction of 3 log units in viable cell count relative to the initial inoculum at all tested concentrations within 1 h, whereas 0.25× MIC (31.25 µg/mL) demonstrated a time-dependent killing property after EGCG treatment in the O1 and O139 clinical strains. However, no bactericidal effect was observed when the bacterial cultures were treated with 0.25× MIC of EGCG in the reference strain (Figure 1c). In contrast, the bacteria performed exponential growth in the absence of EGCG treatment by increasing to approximately 15 log units within 24 h. As a result, it was clear that EGCG was bacteriostatic against *V. cholerae* O1 and O139 MDR strains.

### 2.3. EGCG Disrupts V. cholerae Membrane Permeability

To investigate the mechanism of EGCG on the damaged bacterial cell membrane of the *V. cholerae* reference strain, an effective drug permeability barrier of the gram-negative cell wall, we measured nucleotide and protein leakage, N-phenyl-1-naphthylamine (NPN) uptake, and Rhodamine 123 (Rh123) incorporation, as shown in Figure 2. Bacterial cells were treated with EGCG at various concentrations of 0.25× to 4× MIC (31.25 µg/mL to 500 µg/mL) for 1 h.

The leakage of genetic materials, i.e., DNA, and the amount of protein passing through the bacterial membrane was used to reveal the action of EGCG on the integrity of the membrane. The disruption of the membrane was determined by measuring the cell constituents released; this was done by assessing the absorbance in the supernatant of the bacterial culture treated with EGCG. The results, summarized as the DNA content and protein concentration (Figure 2a,b, respectively), indicated that the released cell constituents increased significantly, in an EGCG concentration-dependent manner.

The outer membrane permeabilization of *V. cholerae* was determined using the NPN uptake assay, where NPN is a neutral hydrophobic fluorescent probe. NPN cannot normally insert into intact bacteria membranes; however, when EGCG disrupts the outer membrane, it gains access to lipid layers in the outer membrane and/or the cytoplasmic membrane, increasing the intensity of its fluorescence emission. As shown in Figure 2c, EGCG slightly permeabilized the outer membrane in a dose-dependent manner, as indicated by an increase in NPN fluorescence. It is important to note that 0.5 mg/mL of EGCG (4× MIC) showed a significant permeabilized cell membrane, equivalent to the positive control Triton X-100. Moreover, we also investigated transmembrane potential activity by staining with Rh123. Considering that Rh123 uptake is proportional to the membrane potential, the results show that EGCG strongly reduces the transmembrane potential in a dose-dependent manner (Figure 2d).

These findings suggest that EGCG may interfere with membrane potential activity, resulting in increased membrane permeability, which causes intracellular ingredient leakage and cell death.

### 2.4. EGCG Altered the Morphological Characterization of V. cholerae

Finally, SEM was used to compare the morphological changes in the appearance of cells with and without 0.5 mg/mL of EGCG exposure. Figure 3 shows SEM images of bacterial cells at ×10,000 and ×20,000 magnifications. The untreated control bacteria had a smooth, compact surface with an intact cell membrane and no surface ruptures (Figure 3a,b). In contrast, after 2 h of exposure to EGCG, the cell was found to be severely disrupted, with membrane corrugations due to withering, wrinkling, and damage, as indicated by the arrow in Figure 3d. Thus, EGCG treatment of bacterial cells typically interferes with the integrity of the cell membranes, resulting in morphological changes that allow for intracellular material leakage, cell membrane shrinkage, and ultimately, cell death.

## 3. Discussion

According to the WHO report, antimicrobial resistance is one of the top 10 global public health threats facing humanity. This resistance is due to the misuse and overuse of drugs, which has resulted in the decreased efficacy of antibiotics [30]. Anti-cholera medications are some of the drugs affected. As a consequence, alternative therapeutic approaches are in high demand. EGCG and green tea catechins have been shown to have a variety of pharmacologically beneficial effects on humans.

Here, our results strongly indicate that EGCG has anti-cholera properties in the MDR *V. cholerae* strains, similar to a previous report where the phytochemical phenolic compounds derived from Piper betle leaf extract (*Piper betle* L.), i.e., piperidine, chlorogenic acid and eugenyl acetate, were all shown to be equally effective against MDR strains of *V. cholerae* [31,32,33]. Moreover, natural compounds had antimicrobial activity against *V. cholerae*, including procyanidins from Guazuma (*Guazuma ulmifolia*) [34], gallate analogues from Daio (*Rhei rhizoma*) [35], apelphenon from apple (*Malus* spp.) [36], procyanidins from hop (*Humulus lupulus*) [37], oil (*diallyl sulphides*) from elephant garlic (*Allium ampleloprasum*) [38], and capsaicin from red chili (*Capsicum annum*) [39,40]. Carvacrol, a major essential oil fraction of oregano (*Origanum vulgare*), inhibited the virulence of *V. cholerae* by inhibiting mucin penetration, adhesion, and the expression of virulence-associated genes (*tcp*A, *ctx*B, *hly*A and *tox*T), resulting in reduced fluid accumulation [41]. On the other hand, cranberry (*Vaccinium macrocarpon*) extract inhibited *V. cholerae* biofilm formation, possibly by modulating the cyclic dimeric guanosine monophosphate (c-di-GMP) level [42]. Furthermore, methanolic extracts of basil (*Ocimum basilicum* L.), nopal cactus (*Opuntia ficus-indica* var. *Villanueva* L.), sweet acacia (*Acacia farnesiana* L.), and white sagebrush (*Artemisia ludoviciana* Nutt.) were found to be the most active against *V. cholerae* via cell membrane disruption [43]. Although several studies have reported anti-cholera infection with natural product extracts, as mentioned above, to the best of our knowledge, this is the first report of antibacterial activity of EGCG against MDR *V. cholerae* strains. According to our findings, EGCG has a synergistic effect with tetracycline. EGCG has long been recognized as a potentially synergistic compound with antibiotics in MDR bacterial clinical strains, such as *Mycobacterium smegmatis* [44], methicillin-resistant *Staphylococcus aureus* (MRSA) [45,46], *Escherichia coli* [47], and *Acinetobacter baumannii* [48].

The modes of action of EGCG and green tea catechins against gram-positive and gram-negative antibacterial activity are classified as follows: (1) inhibition of virulence factors (toxins and extracellular matrix); (2) disruption of cell walls and membranes; (3) inhibition of intracellular enzymes; (4) oxidative stress; (5) DNA damage; and (6) iron chelation [24,49]. The mechanism of action of EGCG is associated with membrane disruption in bacteria cells, such as binding to the bacterial cell membrane, damaging the bacterial cell membrane, inhibiting the ability of bacteria to bind to host cells, inhibiting the ability of bacteria to form biofilms, disrupting bacterial quorum sensing, and interfering with bacterial membrane transporters [20]. However, the mechanism of antibacterial action of EGCG in *V. cholerae* has not yet been reported. Many virulent factors are involved in *V. cholerae* infection, including cholera toxin (haemolysins), toxin coregulated plus (TCP), adhesin factor (ACF), hemagglutination-protease (hap, mucinase), neuramindase, siderophores, and outer membrane proteins, and lipopolysaccharides [50]. Therefore, the modes of action and target sites of EGCG might vary considerably. During bacterial infection, the outer membrane prevents the entry of noxious compounds into the cell, helping them recognize the host and facilitate colonization. This prompted us to speculate that EGCG may influence bacterial membrane permeabilization. As expected, EGCG disrupts the integrity of *V. cholerae* cell membranes by causing intracellular material leakage of both protein and nucleotide, resulting in cell membrane shrinkage and morphological changes that allow for cell death.

However, for future studies, we need to investigate the other modes of action of EGCG, such as virulence gene expression inhibition, as well as the efficacy, safety in animal models and finally, in clinical trials.

## 4. Materials and Methods

### 4.1. Chemicals

EGCG, with more than 98% purity, was kindly provided by Prof. Masami Suganuma at Saitama University (Saitama, Japan). It was extracted from Japanese green tea leaves (*Camellia sinensis* L., O. Kuntze, Theaceae) that were cultured at the Saitama Prefectural Tea Institute in Saitama Prefecture, Japan, as described previously [19,51]. Mueller Hinton Broth (MHB) and Muller Hinton Agar (MHA) were purchased from HiMedia (Mumbai, Maharashtra, India). Rh123, Triton X-100, Glutaraldehyde, and Osmium tetroxide (OsO_4_) were obtained from Sigma Aldrich (St Louis, MO, USA). Resazurin AR (ALPHA CHEMIKA, Mumbai, Maharashtra, India), Tetracycline hydrochloride (PanReac AppliChem, Barcelona, Spain), NPN (TCI, Tokyo, Japan), Bio-Rad DC Protein Assay kit, and bovine serum albumin (BSA) (Bio-Rad Laboratories, Hercules, CA, USA) were used for the experiments.

### 4.2. Bacterial Strains

A total of 45 clinical strains of *V. cholerae* were isolated from stools and rectal swabs, including serogroups O1 and O139, and non-O1/non-O139 strains, originating from Thailand (1983 to 2013). The antimicrobial resistance patterns were characterized in a previous study [12]. *V. cholerae* N16961 was used as a standard reference strain.

### 4.3. Determination of the MIC and the MBC

The MIC values were determined using a 96-well microtiter plate, according to the Clinical and Laboratory Standards Institute (CLSI) guidelines [52], with slight modification, as described in the previous study [53,54]. EGCG was freshly prepared in phosphate buffered saline (PBS) and ethanol at a final concentration of 20%, which was then delicately diluted by using two-fold serial dilution from a concentration of 4.0 mg/mL to 1.95 µg/mL in MHB with 1% NaCl. All wells were inoculated with *V. cholerae* at a final volume of 100 µL of bacterial inoculum (5 × 10^5^ CFU/mL). After incubation for 24 h at 37 °C, 1 mg/mL resazurin was added to all wells (10 µL per well), and incubation then continued for a further 4 h for observation of colour change. The wells with no colour change were appraised as being above the MIC value [55]. On the other hand, the MBC was determined by dropping 10 µL from the wells with concentrations higher than the MIC value directly onto MHA plates and incubating at 37 °C for 24 h. The MBC value was determined when there was no colony growth. Tetracycline, the first line treatment for cholera disease, was used as a positive control [43,56], and media solution was used as a negative control.

### 4.4. Antimicrobial Synergy Testing

The checkerboard assay, which is a two-dimensional array of serial concentrations of test compounds, as previously described [57], was used to determine the potential synergistic activity of EGCG and tetracycline on *V. cholerae* N16961, and also on P48 and 22136, a reference and tetracycline-resistant strain of O1 and O139, respectively. EGCG and tetracycline were prepared in 96-well microtiter plates using two-fold serial dilution based on the MIC of each substance. A final bacterial suspension at 5 × 10^5^ CFU/mL was added to each well. After incubation for 24 h at 37 °C, the synergistic MIC was determined. The observed MIC values were used to calculate the fractional inhibitory concentration (FIC) index; this index allows evaluation of the combined effects of an antibiotic and a compound according to the following formula:FIC (a) = MIC of EGCG in the combination/MIC of EGCG alone;(1)
FIC (b) = MIC of tetracycline in the combination/MIC of tetracycline alone;(2)
FIC index = FIC (a) + FIC (b)(3)

These values were interpreted as follows: for FIC index ≤ 0.5, a synergistic effect; for FIC index > 0.5 and ≤ 4, an additive effect; and for FIC index > 4, an antagonistic effect [58,59].

### 4.5. Time-Kill Kinetics Assay

The killing kinetics of EGCG at 0.25×, 0.5×, 1×, 2× and 4× MIC values were determined using the method described previously [60,61,62], with slight modifications. Different concentrations of EGCG were added to each final volume of 100 μL with 1 × 10^5^ CFU/mL of *V. cholerae* N16961 (reference), P48 (O1), and 22136 (O139) stains grown in MHB containing 1% NaCl and kept at 37 °C. Bacterial growth was monitored over a time-course of 24 h (0, 1, 2, 4, 8, 16, 24 h). A sample without the compound served as a growth control. To evaluate the survival of pandemic strains during the observation period, aliquots of serial dilutions of the bacterial suspensions were determined by a spread plate technique on MHA with 1% NaCl, and the plates were incubated at 37 °C for 24 h to evaluate the viable bacterial colony counts. Data were analysed as killing curves by plotting the log10 CFU/mL versus time (h), and the change in bacterial concentration was determined. The viable bacterial cell count for the time-kill end point determination, i.e., the bactericidal activity, is defined as a reduction of ≥3 log10 CFU/mL relative to the initial inoculum, whereas bacteriostatic activity corresponds to <3 log10 CFU/mL decrease relative to the initial inoculum [63].

### 4.6. Outer Membrane Permeabilization Analysis

#### 4.6.1. Determination of Nucleotide and Protein Leakage

The integrity of the cell membrane can be monitored by the release of cytoplasmic constituents of the cell, using the method described by Lou and others [64], with some modifications. In brief, the *V. cholerae* N16961 cells were cultured overnight at 37 °C. The cells were washed and resuspended at a concentration of 1 × 10^7^ CFU/mL in PBS, pH 7.2. One millilitre of these suspensions was then incubated with EGCG at concentrations of 0.25×, 0.5×, 1×, 2× and 4× MIC at 37 °C for 1 h. After centrifugation, the supernatant samples were immediately filtered through a 0.2 μm organic membrane. To determine the amounts of DNA released from the cytoplasm, the supernatant was used to measure the optical density at 260 nm using a NANO-400A Micro Spectrophotometer (Hangzhou Allsheng Instruments Co.,Ltd. Hangzhou, China). Cell integrity was further examined by determining the release of proteins into the supernatant. The Bradford dye-binding reagent of the Bio-Rad DC Protein Assay kit (Bio-Rad Laboratories, Inc., Hercules, CA, USA) was used to determine the protein amount by measuring the optical density (OD) of the resulting solution at 750 nm within 5 min. The protein quantity of each sample was determined from the equation of the best-fit linear regression obtained from the BSA standard curve. Triton X-100 (0.1%; *v*/*v*) was used as a positive control, while PBS inoculated with the same inoculum was used as a negative control.

#### 4.6.2. Determination of Outer Membrane Disruption

The effect of EGCG on the bacterial outer membrane permeability was determined using an NPN uptake assay [65,66]. Briefly, *V. cholerae* N16961 cells were treated with an appropriate concentration of EGCG at a final volume of 1 mL for 2 h at 37 °C. Then, cell suspensions were washed and resuspended in 0.5% NaCl for 1 mL. The NPN solution in ethanol (100 mM) was added to 200 µL of cells to give a final concentration of 0.75 mM. The background fluorescence was recorded for subtraction using the Cytation 5 Cell Imaging Multi-Mode Reader (BioTek Instruments, Winooski, VT, USA) with an excitation wavelength of 350 nm and an emission wavelength of 420 nm at room temperature. As the outer membrane permeability increased due to the addition of EGCG, NPN incorporated into the membrane resulted in an increase in fluorescence. Triton X-100 (0.1%; *v*/*v*) was used as a positive control. The fluorescence intensity was calculated using the equation:Relative fluorescence intensity (%) = F1/F0 × 100,
where F0 is the fluorescence intensity of untreated cells and F1 is the fluorescence intensity of EGCG-treated cells.

#### 4.6.3. Determination of Cell Membrane Potential

Changes in membrane polarity caused by EGCG were measured through the incorporation of Rh123 [67,68,69]. *V. cholerae* cells were treatment with EGCG at concentrations of 0.25×, 0.5×, 1×, 2×, and 4× MIC at 37 °C for 2 h. The cell suspension was then mixed with a freshly-prepared Rh123 solution (final Rh123 concentration, 5 µg/mL), kept at 37 °C for 10 min, and centrifuged at 1500 rpm for 10 min. Then, cell pellets were diluted in 0.5% NaCl and the fluorescence signal measured at the excitation and emission wavelengths of 480 and 530 nm, respectively. The fluorescence intensity was calculated using the equation mentioned above.

### 4.7. Scanning Electron Microscopy (SEM) Analysis

*V. cholerae* N16961 was treated with EGCG at a concentration of 4× MIC at 37 °C for 2 h. The appropriate treatment was harvested by centrifugation at 5000 rpm for 5 min, washed with PBS, dropped onto a filter membrane of 0.2 µm, and air dried. The samples were fixed using 2.5% (*v*/*v*) glutaraldehyde in PBS at 4 °C overnight. Thereafter, bacterial cells were washed with 0.1 M PO_4_ buffer and re-fixed with 1% OsO_4_ for 1 h. After dehydration with a graded ethanol series (50%, 70%, 90%, and 100%) for 10 min each, the bacterial samples were transferred to absolute ethanol for 20 min. After drying by critical-point drying (CPD), the bacterial sample was then mounted and coated with gold before examination by SEM (JSM 5910 LV, Oxford Instrument, CA, USA) [65].

### 4.8. Statistical Analysis

Values were presented as the mean ± standard deviation (SD) of three independent experiments. The significance of differences between average values of different experimental treatments and controls was assessed by ANOVA, considering that statistical significance was set at a *p* value less than 0.05. When ANOVA revealed significant differences among treatments, post hoc tests were carried out with Dunnett’s Multiple Comparison Test from GraphPad Prism 5.01 (GraphPad Software, Inc., Scotts Valley, La Jolla, CA, USA).

## 5. Conclusions

The findings of the present study highlight the promising role of EGCG as a natural anti-cholera compound against MDR *V. cholerae*. All of the 45 clinical isolates were responsive to EGCG, with MIC and MBC values ranging from 62.5–250 µg/mL and 125–500 µg/mL, respectively. The combination of EGCG and tetracycline was more effective than either treatment alone, with FICIs of 0.009 and 0.018 in O1 and O139 highly drug-resistant representative strains, respectively. According to time-killing kinetics analysis, EGCG had bactericidal activity for MDR *V. cholerae* after 1 h of exposure to at least 62.5 µg/mL EGCG, which is associated with membrane disrupting. As an outcome, EGCG could be promoted as a potential alternative therapeutic agent for MDR *V. cholerae* infections.

## Figures and Tables

**Figure 1 antibiotics-11-00518-f001:**
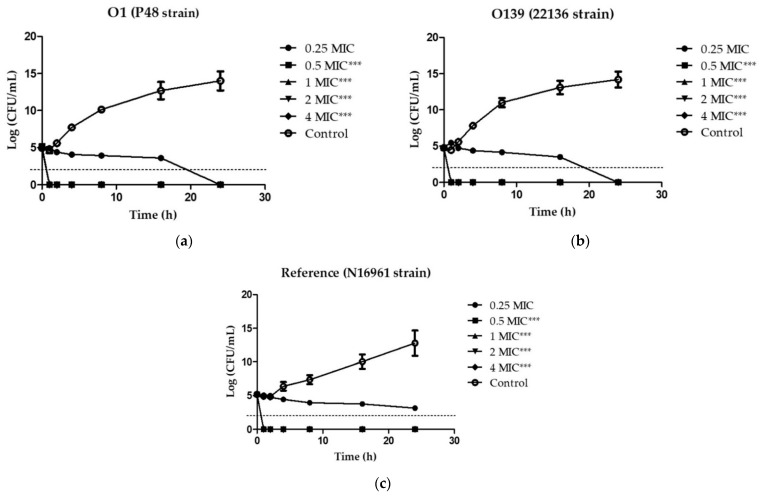
Effect of EGCG on the viability of *V. cholerae*. Time-kill kinetics of *V. cholerae* O1 (**a**), O139 (**b**) and reference strain (**c**) at concentrations of 0.25× to 4× MIC against *V. cholerae* were investigated over a 24 h incubation period at 37 °C. The MIC for EGCG were 125 µg/mL for all strains. MHB was used as the control instead of compound. Samples were taken at 1, 2, 4, 8, 16, and 24 h, to determine viable bacterial numbers. The bactericidal level is indicated by the dashed lines. Significant differences compared to untreated controls at 1 h are indicated by asterisks: *** *p* < 0.001.

**Figure 2 antibiotics-11-00518-f002:**
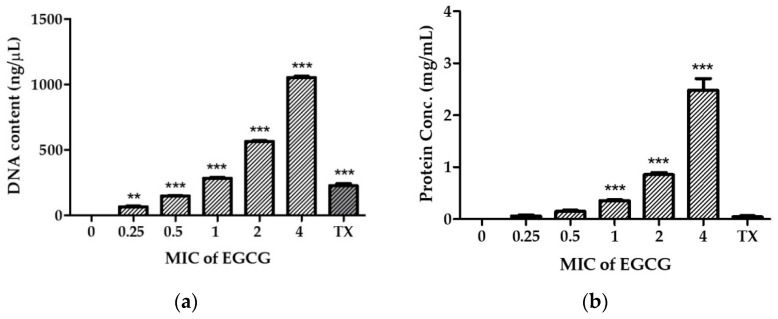
Effect of EGCG on membrane permeability. *V. cholerae* N16961 was treated with EGCG at concentrations of 0.25× MIC to 4× MIC for 2 h at 37 °C. Intracellular leakage of nucleotides (**a**) and proteins (**b**) were measured, and 0.1% Triton X-100 (TX) was used as a positive control. The outer membrane disruption and membrane potential dissipation were investigated in terms of the Relative Fluorescence Intensity (RFI) percentages of NPN (**c**) and Rh123 (**d**). Significant differences compared to untreated controls are indicated by asterisks: * *p* < 0.05, ** *p* < 0.01, *** *p* < 0.001.

**Figure 3 antibiotics-11-00518-f003:**
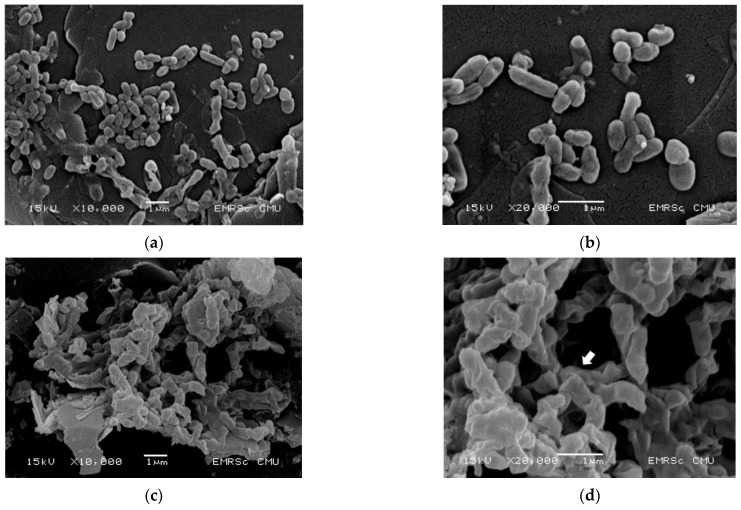
Effect of EGCG on bacterial cell morphology. *V. cholerae* N16961 was treated for 2 h at 37 °C with EGCG 0.5 mg/mL. Scanning electron microscopy (SEM) images at ×10,000 and ×20,000 magnifications were demonstrated: (**a**,**b**) are the controls, (**c**,**d**) are the effective treatments. The cell membrane disruption is represented by the arrow.

**Table 1 antibiotics-11-00518-t001:** MIC and MBC of EGCG for a total of 45 drug resistant *V. cholerae* strains.

No.	Strain	Serogroup/Serotype/Serovar	MIC (µg/mL)	MBC (µg/mL)
1	N16961	*V. cholerae* O1 El Tor Inaba	125	250
2	P33	*V. cholerae* O1 El Tor Ogawa	62.5	125
3	P34	*V. cholerae* O1 El Tor Inaba	62.5	125
4	P35	*V. cholerae* O1 El Tor Inaba	62.5	125
5	P36	*V. cholerae* O1 El Tor Ogawa	62.5	125
6	P38	*V. cholerae* O1 El Tor Inaba	250	500
7	P39	*V. cholerae* O1 El Tor Inaba	125	250
8	P41	*V. cholerae* O1 El Tor Ogawa	62.5	125
9	P42	*V. cholerae* O1 El Tor Inaba	125	250
10	P43	*V. cholerae* O1 El Tor Inaba	62.5	125
11	P44	*V. cholerae* O1 El Tor Inaba	62.5	125
12	P45	*V. cholerae* O1 El Tor Ogawa	125	250
13	P46	*V. cholerae* O1 El Tor Inaba	125	250
14	P47	*V. cholerae* O1 El Tor Ogawa	62.5	125
15	P48	*V. cholerae* O1 El Tor Ogawa	125	250
16	22115	*V. cholerae* O1 El Tor Inaba	125	250
17	22116	*V. cholerae* O1 El Tor Inaba	125	250
18	22118	*V. cholerae* O1 El Tor Inaba	125	250
19	22125	*V. cholerae* O1 El Tor Ogawa	125	250
20	22126	*V. cholerae* O1 El Tor Ogawa	125	250
21	22127	*V. cholerae* O1 El Tor Ogawa	125	250
22	22135	*V. cholerae* O139	62.5	125
23	22136	*V. cholerae* O139	125	250
24	22137	*V. cholerae* O139	125	250
25	22138	*V. cholerae* O139	125	250
26	22144	*V. cholerae* non-O1, non-O139	125	250
27	4053022001	*V. cholerae* O1 El Tor Ogawa	125	250
28	4053023816	*V. cholerae* O1 El Tor Inaba	62.5	125
29	4053023817	*V. cholerae* O1 El Tor Ogawa	125	250
30	4053023818	*V. cholerae* O1 El Tor Inaba	125	250
31	4053023822	*V. cholerae* O1 El Tor Ogawa	62.5	125
32	4053023823	*V. cholerae* O1 El Tor Inaba	62.5	125
33	4053023826	*V. cholerae* O1 El Tor Inaba	62.5	125
34	4053023828	*V. cholerae* O1 El Tor Inaba	125	250
35	4053023829	*V. cholerae* O1 El Tor Inaba	62.5	125
36	4053023830	*V. cholerae* O1 El Tor Ogawa	125	250
37	4053024283	*V. cholerae* O1 El Tor Inaba	125	250
38	4053024290	*V. cholerae* O1 El Tor, Inaba	125	250
39	4053024292	*V. cholerae* O1 El Tor Inaba	125	250
40	4053024293	*V. cholerae* O1 El Tor Inaba	125	250
41	4053024294	*V. cholerae* O1 El Tor Inaba	62.5	125
42	4053024295	*V. cholerae* O1 El Tor Inaba	125	250
43	4053024296	*V. cholerae* O1 El Tor Inaba	125	250
44	4053024297	*V. cholerae* O1 El Tor Inaba	125	250
45	4053024299	*V. cholerae* O1 El Tor Inaba	125	250

Abbreviations: EGCG, (−)-epigallocatechin gallate; MIC, minimum inhibitory concentration; MBC, minimum bactericidal concentration.

**Table 2 antibiotics-11-00518-t002:** Synergistic effect of EGCG in combination with tetracycline against reference and *V. cholerae* MDR strains.

Strain	MIC (µg/mL) of Extracts (a)		FIC (a)	MIC (µg/mL) of Tetracycline (b)		FIC (b)	FICI	Outcome
	Alone	Combination		Alone	Combination			
P48 (O1)	125.0	0.97	0.008	62.5	0.061	0.001	0.009	Synergistic
22136 (O139)	125.0	0.97	0.008	0.78	0.008	0.01	0.018	Synergistic
N16961 (Reference)	125.0	0.97	0.008	3.91	0.004	0.001	0.009	Synergistic

Abbreviations: EGCG, (−)-epigallocatechin gallate; FIC, fractional inhibitory concentration; FICI, fractional inhibitory concentration index; MIC, minimum inhibitory concentration. FIC (a) = MIC of EGCG in the combination/MIC of EGCG alone; FIC (b) = MIC of tetracycline in the combination/MIC of tetracycline alone; FICI = FIC (a) + FIC (b). The values were interpreted as a synergistic effect for FICI ≤ 0.5.

## Data Availability

Not applicable.

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
