# Peer review of "Antimicrobial Activity of the Green Tea Polyphenol (−)-Epigallocatechin-3-Gallate (EGCG) against Clinical Isolates of Multidrug-Resistant *Vibrio cholerae"

_antibiotics, 2022, doi:10.3390/antibiotics11040518_

Round 1
Reviewer 1 Report
This manuscript reported the in vitro antimicrobial properties of EGCG against clinical isolates of multi-drug-resistant (MDR) Vibrio cholerae, and investigated the synergistic effects of EGCG with the antibiotic tetracycline, which was greater than either treatment alone. In addition, the authors also demonstrated the pharmacological mode of action of EGCG might be associated with the disruption of the membrane of the microorganisms. This study might provide a new chance in anti-MDR V. cholerae treatment, however, it lacks substantial interesting work, and I think the manuscript is not suitable for publication in Antibiotics.
- The authors should indicate the activities of the positive and the negative controls for drug resistant cholerae strains in Table 1 and Table 2?
- Except for assessing the synergistic effect of EGCG with the antibiotic tetracycline, what about the effects with other antibiotics, such as fluoroquinolones and azithromycin?
- The EGCG displayed the potential antimicrobial properties in vitro, but what about the in vivo activity? The authors should evaluate the antimicrobial activity using animal models.
- For the mode of action of EGCG, the authors interpret EGCG disrupt the cell membrane cholerae, and ultimately, cell death, if so, what is the mode of action of EGCG against other bacteria and how about the safety in animal?
- For determination of the MIC and MBC, what is the initial concentrations of EGCG?
Author Response
Thank you very much for your stimulating comments. According to your suggestion, we completely revised the manuscript, including our careless mistakes. We hope that the revised manuscript will fulfill your request.
Please see the attachment.

Reviewer 2 Report
This manuscript covers the challenging topic of antimicrobial resistance and combination therapy for cholera. The presentation of results in the manuscript is excellent, and this is a good example of the use of the natural product for the treatment of multi-drug-resistant Vibrio cholerae.
All standard methods were used for the experiments. I found the document interesting for the readers and follow the scope of the journal Antibiotics.
I would recommend the article could be published in Antibiotics, after a minor revision. There are technical errors, I hope the editor will take care of them.
The authors need to address the below-mentioned queries.
- The author could show the structures of (−)-epi-gallocatechin gallate (EGCG), (−)-epigallocatechin (EGC), (−)-epicatechin gal-66 late (ECG), and (−)-epicatechin gallate (EC).
- 2. Tables 1 and 2 needs footnotes, and the author needs to clarify the lines from 96-100. Rearrange the entries 22136 and 22135 in the correct order. The author could use a number in table 1 for better referring.
- The author should mention the controls used in each figure.
- The author could emphasize the criteria selection of P48 V and 22136 V for the synergistic effect of EGCG and tetracycline.
5. The Analysis of Bacterial Killing Kinetics section is not clear and figure 1 is not showing all data of different MIC (Recheck the symbols used in Figure 1). The author did not mention mentions the control used in figure 1.
6. Mention the abbreviation when first used in the text.
7. The author could include the following relevant references.
(i) QIANLING ZHANG, JIN ZHANG, JIAQI ZHANG, DUO XU, YAJUAN LI, YANAN LIU, XIN ZHANG, RUILIN ZHANG, ZUFANG WU, PEIFANG WENG; Antimicrobial Effect of Tea Polyphenols against Foodborne Pathogens: A Review. J Food Prot 1 October 2021; 84 (10): 1801–1808. doi: https://doi.org/10.4315/JFP-21-043
(ii) Maria Daglia, Polyphenols as antimicrobial agents, Current Opinion in Biotechnology, Volume 23, Issue 2, 2012, 174-181,
https://doi.org/10.1016/j.copbio.2011.08.007.
(iii) Food Science and Technology Bulletin: Functional Foods 2 (7) 71–81 DOI: 10.1616/1476-2137.14184.
Author Response
Thank you very much for your favourable and supportive comments. Based on your comments, we revised the manuscript including our mistakes. We hope that these revisions are sufficient for you.
Please see the attachment.

Reviewer 3 Report
The manuscript 'Antimicrobial activity of the green tea polyphenol (−)-epigallocatechin-3-gallate (EGCG) against clinical isolates of multidrug-resistant Vibrio cholerae' is a well-written work that describes the antibacterial activity against Vibrio cholerae exerted by a polyphenol compound present in green tea. The work has a good experimental design and reports interesting findings. It would only need to improve a few minor issues to be suitable for publication in Antibiotics.
Concerns:
1) Please define FIC at its first appearance (line 117).
2) Lines 120-122. A couple of sentences from the MDPI template have been left forgotten. Please delate them.
3) Graphs a, b and c of Figure 1 are quite small and difficult to read. As MDPI has no space limitations, it could be enlarged to improve the readability (and thus, the quality of presentation). For example, two graphs in a first line, the third in a second line.
4) Please avoid leaving big empty spaces as the one of page 7.
5) Material and methods section needs a subsection "Chemicals" or "Materials", in which the different reagents used are cited together with the names of their respective providers, indicating also their city and country. Please specify the city/country of the University of Saitama, and check that all equipments are cited adequately (Company, city, country).
6) The conclusions are very concise, and little informative as they are quite generic. Perhaps they could be completed with, for instance, a brief citation of the assays (MIC determination, synergy...) in which the EGCG showed a more promising activity; and/or by any detail(s) that Authors could considered relevant.
Author Response
Thank you very much for your stimulating comments and friendly encouragement. According to your suggestion, we completely revised the manuscript, including our careless mistakes. We hope that the revised manuscript will fulfill your request.
Please see the attachment.

Round 2
Reviewer 1 Report
The authors have addressed my concerns